# Mineral-Solubilizing Bacteria-Mediated Enzymatic Regulation and Nutrient Acquisition Benefit Cotton’s (*Gossypium hirsutum* L.) Vegetative and Reproductive Growth

**DOI:** 10.3390/microorganisms11040861

**Published:** 2023-03-28

**Authors:** Iqra Ahmad, Maqshoof Ahmad, Azhar Hussain, Muhammad Zahid Mumtaz, Ghulam Hassan Abbasi, Farheen Nazli, Lisa Pataczek, Hayssam M. Ali

**Affiliations:** 1Department of Soil Science, The Islamia University of Bahawalpur, Bahawalpur 63100, Pakistan; 2Institute of Molecular Biology and Biotechnology, The University of Lahore, Lahore 54000, Pakistan; 3Institute of Agro-Industry and Environment, The Islamia University of Bahawalpur, Bahawalpur 63100, Pakistan; 4Institute of Landscape and Plant Ecology, University of Hohenheim, Ottilie-Zeller-Weg 2, 70599 Stuttgart, Germany; 5Department of Botany and Microbiology, College of Science, King Saud University, Riyadh 11451, Saudi Arabia

**Keywords:** *Bacillus* spp., cotton, enzymatic activity, nutrient uptake, *Paenibacillus* spp., phosphate solubilizing bacteria, zinc solubilizing bacteria

## Abstract

Many farmers’ incomes in developing countries depend on the cultivation of major crops grown in arid and semi-arid regions. The agricultural productivity of arid and semi-arid areas primarily relies on chemical fertilizers. The effectiveness of chemical fertilizers needs to improve by integration with other sources of nutrients. Plant growth-promoting bacteria can solubilize nutrients, increase plant nutrient uptake, and supplement chemical fertilizers. A pot experiment evaluated the promising plant growth-promoting bacterial strain’s effectiveness in promoting cotton growth, antioxidant enzymes, yield, and nutrient uptake. Two phosphate solubilizing bacterial strains (*Bacillus subtilis* IA6 and *Paenibacillus polymyxa* IA7) and two zinc solubilizing bacterial strains (*Bacillus* sp. IA7 and *Bacillus aryabhattai* IA20) were coated on cotton seeds in a single as well as co-inoculation treatments. These treatments were compared with uninoculated controls in the presence and absence of recommended chemical fertilizer doses. The results showed the co-inoculation combination of *Paenibacillus polymyxa* IA7 and *Bacillus aryabhattai* IA20 significantly increased the number of bolls, seed cotton yield, lint yield, and antioxidants activities, including superoxide dismutase, guaiacol peroxidase, catalase, and peroxidase. Co-inoculation combination of *Bacillus subtilis* IA6 and *Bacillus* sp. IA16 promoted growth attributes, including shoot length, root length, shoot fresh weight, and root fresh weight. This co-inoculation combination also increased soil nutrient content. At the same time, *Paenibacillus polymyxa* IA7 + *Bacillus aryabhattai* IA20 increased nutrient uptake by plant shoots and roots compared.

## 1. Introduction

Cotton is a fiber crop grown in arid and semi-arid regions. These regions have significant issues with poor nutrient availability, especially those with low diffusion coefficients [1,2,3]. Pakistan ranked fifth in producing cotton, second for export, and seventh for cloth production globally. Recent estimates revealed cotton cultivation on 2079 thousand hectares with an annual output of 7.064 million bales [4]. The agricultural productivity of these regions mainly depends upon the utilization of chemicals, as it is considered an integral part of farming systems.

Phosphorus (P) is an essential macronutrient for plant growth and development. It plays a critical role in photosynthesis, root growth and development, stem strengthening, seeds and flower formation, crop maturity, and quality production. It is involved in energy generation, storage, transfer reactions, cell division, and cell enlargement in plants [5,6,7]. P plays a significant role in biological processes, but in arid and semi-arid regions, the low concentration of P in soil results in a lower yield of crops, particularly wheat, cotton, rice, and soybean [8]. In cotton, P helps in the early developmental phase [9]. Approximately 5.7 billion ha^−1^ (40%) of cultivated areas are deficient in P worldwide [8]. While 80% of P applied through fertilizer remains inaccessible to plants [10] because of fixation with other ions such as oxides, hydroxides of iron, and aluminum [11] and with calcium [7]. Similarly, zinc (Zn) is a micronutrient but equally essential as P. It is the structural part of enzymes. It acts as a co-factor of various enzymes that perform functions in photosynthesis, carbohydrate metabolism, and the formation of starch from sugars [12]. Moreover, Zn also plays an essential role in pollen formation, protein and auxin metabolism, and the integrity of biological membranes [13,14]. Nowadays, the whole world is facing Zn deficiency in agricultural soils. Approximately 70% of Pakistani farm soils are Zn deficient [15]. Such grounds are for calcareous alkaline soil as Zn becomes unavailable to plants. Zn-containing fertilizers are needed to meet the crop’s needs [16,17,18].

Chemical fertilizers are applied to cope with this problem, but these have the disadvantages of immobilization and runoff in erosion-prone areas resulting in environmental pollution and a high production cost [19,20]. Moreover, naturally occurring minerals decrease over time because of their overuse and shortage of resources [21]. Chemical fertilizer’s efficiency is needed by integrating with natural sources of nutrients. Biofertilizers composed of plant growth-promoting bacteria (PGPB) application could be an excellent supplement to chemical fertilizers. PGPBs populate the rhizosphere and compete with other microorganisms for food and survival [22]. These bacteria increase nutrient concentration by nutrient solubilization in soil and nutrient availability for plants [23,24,25]. PGPB application in agriculture is increasing; however, their quality and sustainability are not up to mark [26]. Several studies have reported the PGPB’s role in better crop growth and yield [27,28]. Naseer et al. [17] reported increased rice growth by applying Zn solubilizing *Bacillus* strains. Majeed et al. [23] also reported a rise in wheat growth treated with native PGPB.

Similarly, integrated application of phosphate solubilizing *Bacillus* IA6 and Zn solubilizing *Bacillus* sp. IA16 showed a promising increase in cotton growth [28]. Such PGPB can also improve vegetative and reproductive crop growth, reduces dependence on chemical fertilizers, and plays an essential role in environmental protection [18,29]. For this purpose, the current study was performed to improve cotton crop growth, yield, and chemical attributes by sole and co-inoculation combinations of phosphate solubilizing bacteria (PSB) and Zn solubilizing bacteria (ZSB) strains in the presence and absence of chemical fertilizers. Cotton seeds treated with sole and co-inoculation combinations of two PSB strains (*B. subtilis* IA6 and *P. polymyxa* IA7) and two ZSB strains (*Bacillus* sp. IA7 and *B. aryabhattai* IA20) were tested in a pot trial. The co-inoculation combination of *P. polymyxa* IA7 and *B. aryabhattai* IA20 reported a maximum increase in antioxidant enzyme activities, reproductive growth, and nutrient concentration in cotton root and shoot tissues. While the co-inoculation combination of *B. subtilis* IA6 and *Bacillus* sp. IA16 demonstrated the best performance regarding increased vegetative growth and nutrient availability in soil. 

## 2. Materials and Methods

### 2.1. Culturing of PSB and ZSB Strains

Two PSB strains [*Bacillus subtilis* IA6 (MN005922) and *Paenibacillus polymyxa* IA7 (MN005923)] and two ZSB strains [*Bacillus* sp. IA16 (MN005924) and *Bacillus aryabhattai* IA20 (MN005925)] were obtained from the Laboratory of Soil Microbiology and Biotechnology, Department of Soil Science (DSS), Faculty of Agriculture and Environmental Sciences (FAES), the Islamia University of Bahawalpur (IUB), Pakistan. Previously, these strains reported in vitro solubilization of phosphate and zinc and plant growth-promoting characteristics, e.g., production of siderophores, hydrogen cyanide, ammonia, and exopolysaccharides. These strains also reported cotton growth promotion under controlled condition jar trial [28]. The tested strains were grown in Dworkin and Foster (DF) salt minimal broth [30] for two days at 28 ± 1 °C under shaking (100 rpm) conditions. The cultures with an optical density of 0.50 at 600 nm wavelength were used to coat cotton seeds and slurry-based carrier materials, including non-sterile peat and sugar solution, following the recipe reported by Mumtaz et al. [30]. 

### 2.2. Soil Analysis

The soil used in the current pot experiment was obtained from a farmer’s field. Soil physicochemical characteristics were determined by adapting the methods described in Handbook 60 [31]. Soil samples were collected before sowing using a drill from 0–15 cm depth. Soil textural class was determined by adopting the methods of Moodie et al. [32]. According to this method, a soil solution composed of soil (50 g), (NaPO_3_)_6_ (1% solution), and distilled water (250 mL) was incubated overnight. The solution was transferred to a 1 L graduated cylinder, and distilled water was added up to the mark. The mixture was stirred homogeneously and read through the Bouyoucos hydrometer (Analytikia, Thessaloniki, Greece). Reading was plotted on a soil texture triangle, and soil textural class was estimated [31]. 

The soil-saturated paste was used to estimate the saturation percentage, pH, and electrical conductivity (EC). It was prepared using 250 g of soil, and the saturation percentage was assessed by determining the weight of the saturated paste and oven-dry weight. The pH of soil-saturated paste was determined through calibrated digital pH meter (model: Kent Eil 7015). The extract of soil-saturated paste was obtained through a vacuum pump, and the EC of the soil extract was determined by calibrated digital Jenway conductivity meter [31]. To determine organic matter contents in soil, a mixture composed of soil (1 g), K_2_Cr_2_O_7_ (10 mL of 1 N), H_2_SO_4_ (20 mL), distilled water (150 mL), and FeSO_4_ (25 mL of 0.5 N) was titrated against KMnO₄ (0.1 N) up to pink endpoint [32].

To determine total nitrogen (N) contents, the soil sample was digested with concentrated H_2_SO_4_ and a catalyst mixture (K_2_SO_4_-CuSO_4_.5H_2_O-Se in 100:10:1 *w*/*w* ratio) at 370 °C. The cooled mixture was distilled with saturated H_3_BO_3_ and titrated with dilute H_2_SO_4_ to pH 5.0. Further, as described by Jackson [33], total nitrogen (N) was determined through the Kjeldahl apparatus (VELP Scientifica, Usmate Velate, MB, Italy). To estimate available P in soil, a soil sample (2 g) was mixed with 60% perchloric acid (30 mL) along with five pumice-boiling granules (Sigma-Aldrich, Burghausen, Germany) and heated up to 180 °C until the appearance of white soil material. The digest was cooled at room temperature, filtered through Whatman filter paper, and diluted with distilled water up to 250 mL. The soil digest (5 mL) was reacted with 10 mL of ammonium-vanadomolybdate reagent and read at 470 nm. The available P in soil was estimated by plotting the standard curve prepared value of the stock solution of 2, 4, 6, 8, and 10 ppm KH_2_PO_4_ [34]. The soil digest was read through a flame photometer (BWB-XP, BWP Technologies, Berkshire UK) for the determination of potassium (K) [31] and through atomic absorption spectrophotometer (240FS AA, Agilent Technologies, Mulgrave VIC, Australia) for the determination of Zn and iron (Fe) in soil [35]. The physicochemical properties of the soil used in a pot trial are given in Table 1.

### 2.3. Pot Experiment

A pot experiment was conducted in the wire house of the Department of Soil Science, FA&ES, IUB, Bahawalpur, Pakistan, at a latitude of 29.40N, longitude: 71.68E, and 116 m elevation above sea level. Cotton seeds of cultivar IUB-3 were coated with PSB, ZSB strains, and carrier materials in sole inoculation (IA6, IA7, IA16, and IA20) and co-inoculation combinations (IA6 + IA16, IA6 + IA20, IA7 + IA16, and IA7 + IA20). For combined application, strain broth cultures were maintained at a 1:1 ratio by applying bacterial liquid cultures of 0.70 optical density at 600 nm. For seed coating, the wet slurry was prepared with sterilized peat, broth culture, and sterilized sugar solution (10%) in a 5:4:1 ratio and mixed with cotton seeds to obtain sufficient coating. Two treatments, an uninoculated control without the recommended doses of nitrogen, phosphorus, and potassium (NPK) chemical fertilizers (C1: absolute control) and an uninoculated control with the recommended doses of NPK chemical fertilizers (C2: recommended NPK doses), were also run to compare the effect of the treatment. The recommended doses of NPK (200: 90: 75 kg ha^−1^; Agriculture Department, Punjab, Pakistan) were applied before sowing in the form of urea, diammonium phosphate, and sulfate of potash. For control treatments (C1 and C2), seeds were coated with all particulars except the strain’s cultures. 

The earthen pots were loaded with 12 kg of dry (dried under shade for two weeks) and sieved (through 2 mm iron mesh) soil. Ten cotton seeds from each treatment were sown in the pots. The pots were arranged in a completely randomized design (CRD) with six replications. The trial was carried out under natural climatic conditions. The pots were regularly irrigated with good-quality water to fulfill the crop’s requirements [36]. The plant’s population was maintained at three plants in each pot by thinning after germination. Antioxidant activities were estimated at the flowering stage. The growth and yield parameters were determined at harvesting. After harvesting the crop, soil and plant samples were collected to assess plant nutrient uptake.

### 2.4. Determination of Antioxidative Enzymes

The third top leaf from each pot was collected at the flowering time to determine antioxidant enzyme activities. The fresh leaf samples (1.0 g) were crushed and soaked in 3 mL ice-cold phosphate buffer solution (PBS-100 mM; prepared through disodium hydrogen phosphate (16.385 g) and sodium dihydrogen phosphate (0.663 g) in one liter of distilled water). The mixture was homogenized through a vortex and centrifuged at 16,000× *g* for 15 min at 4 °C. The supernatant was used as the source of enzymes. Superoxide dismutase (SOD) activity was determined using enzymatic extract (50 µL) and reaction solution of *p*-nitro blue tetrazolium (75 µM L^−1^), riboflavin (20 µM L^−1^), ethylene diamine tetra acetic acid (EDTA; 100 µM L^−1^), methionine (13 µM L^−1^), and PBS (50 mM). The absorbance of the reaction mixture was read at 560 nm through a mass spectrophotometer (Model G6860A, Agilent Technologies Cary 60 UV-Vis, Mulgrave VIC, Australia). The control group was also maintained by adding all the mentioned reaction solutions except enzymatic extract, and the reaction mixture was kept in the dark before reading absorbance [37]. 

Guaiacol peroxidase activity (GPX) was determined using the reagent mixture of 9 mM guaiacol, 50 mM phosphate buffer, and 19 mM H_2_O_2_ [38]. The kinetic evolution of absorbance at 470 nm was measured for 1 min. GPX activity was calculated using the extinction coefficient (26.6 mM^−1^ cm^−1^ at 470 nm). One unit of peroxidase was defined as the amount of enzyme that caused the formation of 1 mM of tetra guaiacol per minute. To determine catalase activity (CAT) in cotton leaves, enzyme extract (50 µL) was mixed with a 3 mL solution composed of H_2_O_2_ (300 mM) and PBS buffer (50 mM). The samples were gently shaken, and absorbance was measured using a mass spectrophotometer at 240 nm wavelength [39,40]. For peroxidase (POX) activity, crude enzymes extract (25 µL) was mixed with 0.01 M pyrogallol (1 mL), 0.1 M phosphate buffer (2 mL), and 0.005 M H_2_O_2_ (1 mL). After 10 min. of incubation, the reaction was stopped by adding 0.5 mL H_2_SO_4_ (5%), and absorbance was recorded at 480 nm [38].

### 2.5. Growth and Yield Attributes

The shoot length and root length were measured through a meter rod. The shoot and root fresh weights were determined using a portable weight balance. These shoot and root samples were dried under shade for four weeks at 25 °C, and shoot and root dry weights were obtained through weight balance. The sympodial branches and number bolls were counted manually, and the dry cotton from the open boll was manually picked and weighed through weight balance. Further, dry cotton and seeds were separated and weighted to represent lint yield. The lint percentage of seed cotton was determined by dividing lint weight by total seed cotton weight. 

### 2.6. Determination of Nutrients Concentration in Soils, Roots, and Shoots

The root and shoot parts from uprooted plants were separated, and soil samples were collected from each treatment. Before digestion, plant samples, i.e., shoots containing leaves and stems, roots, and soils, were dried in an oven at 67 °C till constant weight and ground. The plant samples were digested using H_2_SO_4_ and H_2_O_2_ following Wolf’s [41] procedures. According to this method, plant samples were taken in a 250 mL conical flask, and concentrated H_2_SO_4_ was added. The tubes were incubated overnight at room temperature. Further, 30% H_2_O_2_ was added to tubes and heated to 350 °C on a hot plate. More H_2_O_2_ was added to the cooled mixture to get the white transparent plant digest. The digest was diluted and filtered with Whatman filter paper. 

The N concentration in plant digest was determined by using the Kjeldahl distillation and digestion method. P concentrations were measured by adding a color-developing reagent (Barton reagent), and P concentration was calculated by plotting the standard curve [42]. K concentration was determined by using a flame photometer. The Zn and Fe concentrations in plant digest were determined by reading samples on Atomic Absorption Spectrophotometry. 

### 2.7. Statistical Analysis

Recorded data were statistically analyzed through Statistix 8.1 (Analytical Software, Tallahassee, FL, USA) using a complete randomized design (CRD). The one-way analysis of variance (ANOVA) was applied, and treatment means were compared using the least significant difference method (LSD) at the 5% probability level [43,44].

## 3. Results

### 3.1. Shoot Growth

The PSB and ZSB strains significantly enhanced the shoot length, shoot fresh weight, and dry weight of cotton plants (Figure 1A and Figure 2A,B, respectively). The recommended NPK doses increased shoot length, shoot fresh weight, and shoot dry weight up to 2.3%, 3.9%, and 5.4%, respectively, over the absolute control. In the case of sole inoculations, a maximum increase in shoot length was reported by *Bacillus subtilis* IA6, with an increase of up to 10% over the absolute control and 7.5% over the recommended NPK doses. In contrast, *Bacillus aryabhattai* IA20 showed maximum improvement in shoot fresh weight up to 14.5% over the absolute control and 10% over the recommended NPK doses. It also showed higher shoot dry weight with an increase of 15% compared to the absolute control and 9.2% compared to the recommended NPK doses. Among co-inoculations, *Bacillus subtilis* IA6 + *Bacillus* sp. IA16 significantly increased shoot length up to 19.7% and 17.0% and shoot fresh weight up to 19.5% and 14.9% compared to the absolute control and recommended NPK doses. Maximum shoot dry weight was obtained from co-inoculation combination *Paenibacillus polymyxa* IA7 + *Bacillus* sp. IA16 increased to 18.9% over the absolute control and 12.8% over the recommended NPK doses.

### 3.2. Root Growth

The recommended NPK doses increased root length, root fresh weight, and root dry weight by up to 6.0%, 6.8%, and 6.0%, respectively, and remained non-significant over the absolute control (Figure 1A and Figure 2A,B, respectively). The sole inoculation with *Bacillus aryabhattai* IA20 significantly increased root length and dry weight by 11.6% and 10.0%, respectively, compared to the absolute control and 5.3% and 3.8%, respectively, compared to the recommended NPK doses. The sole inoculation with *Bacillus subtilis* IA6 followed by *Paenibacillus polymyxa* IA7 reported better root fresh weight with 9.7% and 9.5%, respectively, over the absolute control and 2.7% and 2.5%, respectively, over the recommended NPK doses. The co-inoculation combination of *Bacillus subtilis* IA6 + *Bacillus* sp. IA16 reported the highest root length and root dry weight with 18.0% and 18.6%, over the absolute control and up to 11.4% and 11.6%, respectively, compared to the recommended NPK doses. The co-inoculation with *Paenibacillus polymyxa* IA7 + *Bacillus* sp. IA16 reported maximum root fresh weight, increasing by up to 20.3% (over the absolute control) and 12.6% (over the recommended NPK doses), followed by *Bacillus subtilis* IA6 + *Bacillus* sp. IA16 demonstrated an 18.6% and an 11.1% increase in root fresh weight compared to the absolute control and recommended NPK doses.

### 3.3. Reproductive Growth

The recommended NPK doses increased the number of bolls and the number of sympodial branches by 2.5% and 11.6%, respectively, over the absolute control and remained non-significant to each other (Figure 1B). Similarly, single boll weight was 6.0% higher in the recommended NPK than the absolute control and significantly differed from each other (Table 2). The co-inoculation with *Paenibacillus polymyxa* IA7 + *Bacillus aryabhattai* IA20 significantly increased the number of bolls by 11.9% compared to the absolute control and 9.1% compared to the recommended NPK doses. It also enhanced the number of sympodial branches up to 29.8% compared with absolute control and 16.2% compared to the recommended NPK doses. The single boll weight with an increase of 13.4% and 7.0% and the number of monopodial branches with a rise of 22.6% and 15.3% were observed from *Paenibacillus polymyxa* IA7 + *Bacillus* sp. IA16 compared to the absolute control and recommended NPK doses, respectively. Sole inoculation with *Bacillus aryabhattai* IA20 increased the number of bolls, single boll weight, and monopodial branches by 5.1%, 11.1%, and 17.3% compared to the absolute control and 2.5%, 4.8%, and 10.2% compared to the recommended NPK doses. *Bacillus* sp. IA16 significantly increased the number of sympodial branches to a 28.5% increase over the absolute control and 15.1% over the recommended NPK doses.

### 3.4. Yield Attributes

The cotton seed yield, lint yield, and lint percentage were up to 8.7%, 11.6%, and 0.2% higher, respectively, compared to the absolute control (Table 2). Among sole inoculations, *Bacillus aryabhattai* IA20 significantly increased cotton seed and lint yield by up to 16.7% and 19.8%, respectively, compared to the absolute control and up to 7.4% and 7.7%, respectively, compared to the recommended NPK doses. However, a maximum lint percentage with an increase of 1.6% over the absolute control and 1.3% over the recommended NPK doses was reported by *Paenibacillus polymyxa* IA7 treated plants. *Bacillus aryabhattai* IA20 also showed a better lint percentage, increasing to 0.97% and 0.65 over the absolute control and recommended NPK doses. The most effective increase in cotton seed yield and lint yield was shown by *Paenibacillus polymyxa* IA7 + *Bacillus aryabhattai* IA20, with an increase of up to 24.8% compared to the absolute control and 23.9% compared to the recommended NPK doses. This treatment was followed by *Bacillus subtilis* IA6 + *Bacillus aryabhattai* IA20, which reported 14.0% over the absolute control and 11.8% over the recommended NPK doses. The co-inoculation treatment *Paenibacillus polymyxa* IA7 + *Bacillus* sp. IA16 showed maximum improvement in lint percentage with an increase of 2.4% and 2.1% compared with the absolute control and recommended NPK doses. This treatment was followed by *Paenibacillus polymyxa* IA7 + *Bacillus aryabhattai* IA20, having increased lint percentage up to 1.2% and 1.0% over the absolute control and recommended NPK doses, respectively.

### 3.5. Antioxidant Activity

Co-inoculation significantly increased antioxidant activity compared to the absolute control and recommended NPK doses. *Paenibacillus polymyxa* IA7 + *Bacillus aryabhattai* IA20 followed by *Bacillus subtilis* IA6 + *Bacillus* sp. IA16 showed significant maximum SOD activity (Figure 3A). These co-inoculation combinations increased SOD activity by 11.2% and 10.7%, respectively, compared with the absolute control, and 7.2% and 6.7%, respectively, compared with the recommended NPK doses. Among sole inoculations, the maximum rise of 5.3% and 1.5% in SOD activity was reported by *Bacillus subtilis* IA6 compared to the absolute control and recommended NPK doses. A significant increase in GPX activity up to 18.3% over the absolute control and 12.5% over the recommended NPK doses was observed from *Paenibacillus polymyxa* IA7 + *Bacillus* sp. IA16 (Figure 3B). Among sole inoculations, the maximum GPX activity was reported by *Bacillus subtilis* IA6 with a rise of 9.9% compared to the absolute control and 4.5% compared to the recommended NPK doses. *Paenibacillus polymyxa* IA7 + *Bacillus aryabhattai* IA20 showed a significant increase in CAT activity at 8.3% and 5.0%, followed by *Bacillus subtilis* IA6 + *Bacillus* sp. IA16 showed 7.2% and 4.0% higher CAT activity than the absolute control and recommended NPK doses, respectively (Figure 3C). Among sole inoculations, significant CAT activity was observed from *Bacillus aryabhattai* IA20 with a rise of 6.5% and 3.3% compared with the absolute control and recommended NPK doses. A significant rise in POX activity was reported by co-inoculation with *Paenibacillus polymyxa* IA7 + *Bacillus* sp. IA16 by 12.9% compared to the absolute control and 10.2% compared to the recommended NPK doses (Figure 3D). Sole inoculation with *Bacillus aryabhattai* IA20 also promoted POX activity up to 10.6% over the absolute control and 7.9% over the recommended NPK doses. 

### 3.6. Macronutrient Dynamics 

Data regarding the effect of inoculation with PSB and ZSB strains on N content in soil (Appendix A), roots, and shoots of cotton are given in Table 3 and Table 4. Co-inoculation treatments were more efficient in improving macronutrient contents in soil than sole application. Co-inoculation with *Paenibacillus polymyxa* IA7 + *Bacillus aryabhattai* IA20 enhanced total soil N content by 11.1% and 6.2% compared to the absolute control and recommended NPK doses, respectively. This treatment was followed by *Bacillus subtilis* IA6 + *Bacillus* sp. IA16 causes an increase in total soil N contents by 10.4% and 5.5% over the absolute control and recommended NPK doses, respectively. Among sole inoculations, a significant increase of 8.3% and 3.4% in total soil N was caused equally by both *Bacillus subtilis* IA6 and *Bacillus* sp. IA16 as compared to the absolute control and recommended NPK doses, respectively.

Co-inoculation with *Bacillus subtilis* IA6 + *Bacillus* sp. IA16 significantly increased N content in cotton roots by 9.0% and 5.0% compared to the absolute control and recommended NPK doses. After that, *Paenibacillus polymyxa* IA7 + *Bacillus aryabhattai* IA20 showed significant improvement up to 8.5% and 4.6% compared to the absolute control and recommended NPK doses. Among sole inoculations, *Bacillus* sp. IA16 significantly increased N content in the root by 5.6% over the absolute control and 1.7% compared to recommended NPK doses. A significant increase in shoot N content was reported by *Bacillus subtilis* IA6 + *Bacillus* sp. IA16 raised shoot N contents by 8.3% and 4.2% over the absolute control and recommended NPK doses. *Paenibacillus polymyxa* IA7 + *Bacillus aryabhattai* IA20 enhanced shoot N content up to 7.2% and 3.2% over the absolute control and recommended NPK doses. Among sole inoculations, the most significant N content in the shoot was reported by *Bacillus aryabhattai* IA20, with an increase of 4.8% compared with the absolute control and 0.8% compared with the uninoculated control.

Co-inoculation treatments were superior in promoting P contents in soil, roots, and shoots of cotton than sole inoculations (Appendix A; Table 3 and Table 4). The maximum improvement in soil P concentration was developed by *Bacillus subtilis* IA6 + *Bacillus* sp. IA16 followed by *Paenibacillus polymyxa* IA7 + *Bacillus* sp. IA16. These treatments increased soil P concentration by 13.3% and 12.7%, respectively, compared to the absolute control, while 9.3% and 8.7%, respectively, increased soil P concentration was observed compared with the recommended NPK doses. *Bacillus* sp. IA16 increased soil P contents to 9.3% and 5.4% over the absolute control and recommended NPK doses. *Bacillus aryabhattai* IA20 also showed a better increase of up to 8.7% and 4.8% in soil P content over the absolute control and recommended NPK doses.

The combination of *Paenibacillus polymyxa* IA7 + *Bacillus aryabhattai* IA20 followed by *Bacillus subtilis* IA6 + *Bacillus* sp. IA16 demonstrated higher root P contents with an increase of up to 11.4% and 10.2%, respectively, compared to the absolute control and 6.5% and 5.2%, respectively, compared to the recommended NPK doses. Among sole inoculations, *Paenibacillus polymyxa* IA7 reported maximum P content in the root, increasing up to 8.5% and 3.6% over the absolute control and recommended NPK doses. The most significant enhancement in P content in the shoot was shown by *Paenibacillus polymyxa* IA7 + *Bacillus* sp. IA16 followed by *Bacillus subtilis* IA6 + *Bacillus* sp. IA16 with an increase of 10.8% and 9.7%, respectively, compared with the absolute control and 7.2% and 6.0%, respectively, compared with the recommended NPK doses. All the sole treatments enhanced P content in cotton shoots non-significantly except *Paenibacillus polymyxa* IA7, which demonstrated an increase of up to 8.0% compared with the absolute control and 4.4% compared with the recommended NPK doses. 

*Bacillus subtilis* IA6 + *Bacillus* sp. IA16 increased soil K content with 11.3% and 8.4% enhancement compared to the absolute control and recommended NPK doses (Appendix A). Sole inoculation with *Bacillus aryabhattai* IA20 followed by *Paenibacillus polymyxa* IA7 non-significantly improves K content in soil up to 6.1% and 5.9%, respectively, over the absolute control and up to 3.3% and 3.1%, respectively, compared with the recommended NPK doses. The combination of *Paenibacillus polymyxa* IA7 + *Bacillus* sp. IA16 showed significantly higher root K content, increasing 9.7% over the absolute control and 7.4% over the recommended NPK doses (Table 3). *Paenibacillus polymyxa* IA7 + *Bacillus aryabhattai* IA20 increased root K contents up to 9.5% compared to the absolute control and up to 7.2% over the recommended NPK doses. Sole inoculation with *Bacillus aryabhattai* IA20 significantly enhanced K content with an increase of 7.7% compared to the absolute control and 5.5% compared to the recommended NPK doses. Maximum improvement in shoot K contents was observed from *Paenibacillus polymyxa* IA7 + *Bacillus* sp. IA16 (9.5% higher than the absolute control and 6.1% higher than the recommended NPK doses) followed by *Paenibacillus polymyxa* IA7 + *Bacillus aryabhattai* IA20 (8.5% higher than the absolute control and 5.1% higher than the recommended NPK doses) (Table 4). Among sole inoculations, *Bacillus aryabhattai* IA20 (7.4% and 4.1% higher than the absolute control and recommended NPK doses) followed by *Bacillus* sp. IA16 (4.5% and 1.2% higher than the absolute control and recommended NPK doses) reported a better increase in shoot K contents.

### 3.7. Micronutrient Dynamics 

A significant improvement in shoot Zn content was shown by *Paenibacillus polymyxa* IA7 + *Bacillus aryabhattai* IA20 with an increase of up to 14.3% compared to the absolute control and 12.8% compared to the recommended NPK doses (Figure 4A). Among sole inoculations, a better increase in shoot Zn concentration was shown by *Bacillus aryabhattai* IA20, followed by *Paenibacillus polymyxa* IA7 with a rise of 7.6% and 5.1%, respectively, compared to the absolute control and up to 6.2% and 3.7%, respectively, compared to the recommended NPK doses. The Zn solubilizing strain *Bacillus* sp. IA16 showed the maximum root Zn content, an increase of 5.1% over the absolute control and 2.1% over the recommended NPK doses (Figure 4B).

The combined application of *Paenibacillus polymyxa* IA7 + *Bacillus aryabhattai* IA20 also showed better improvement in root Zn content with an increase of up to 14.8% and 11.5% compared to the absolute control and recommended NPK doses. The combination of *Bacillus subtilis* IA6 + *Bacillus* sp. IA16 reported the maximum enhancement in soil Zn content, followed by *Bacillus subtilis* IA6 + *Bacillus aryabhattai* IA20 (Figure 4C). These combinations increased soil Zn content to 12.3% and 10.9%, respectively, compared to the absolute control and 10.8% and 9.5%, respectively, compared to the recommended NPK doses. *Bacillus aryabhattai* IA20 significantly increased soil Zn content up to 6.4% and 5.0% compared to the absolute control and recommended NPK doses. 

*Bacillus aryabhattai* IA20 enhanced shoot Fe content with a rise of 8.1% compared to the absolute control and 7.0% compared to the recommended NPK doses (Figure 5A). Co-inoculation of strains exhibited effective results in improving Fe content in the shoot. The combined application of *Bacillus subtilis* IA6 + *Bacillus* sp. showed a significant improvement in shoot Fe content. IA16 followed by *Paenibacillus polymyxa* IA7 + *Bacillus aryabhattai* IA20. These treatments enhanced shoot Fe content by 11.2% and 10.3%, respectively, compared to the absolute control and 10.2% and 9.2%, respectively, over the recommended NPK doses. *Bacillus* sp. IA16 and *Paenibacillus polymyxa* IA7 increased root Fe content equally up to 8.4% and 3.1% compared to the absolute control and recommended NPK doses (Figure 5B). Among co-inoculation treatments, *Paenibacillus polymyxa* IA7 + *Bacillus aryabhattai* IA20 showed a maximum increase in root Fe content by 10.9% and 5.5%, followed by *Bacillus subtilis* IA6 + *Bacillus* sp. IA16 showed 10.5% and 5.1% higher root Fe content over the absolute control and recommended NPK doses. Maximum soil Fe content was obtained from a combination of *Bacillus subtilis* IA6 + *Bacillus aryabhattai* IA20, causing an increase of 10.9% compared to the absolute control and 6.3% compared to the recommended NPK doses (Figure 5C). *Bacillus subtilis* IA6 also showed better improvement in soil Fe content with an enhancement of 7.3% and 2.9% over the absolute control and recommended NPK doses.

## 4. Discussion

Despite the abundance of P in agricultural soils, most of it remains insoluble and becomes unavailable to plants [45]. P makes complexes with aluminum, Fe, and hydroxides (in acidic soils) and with calcium (in alkaline soils) [46]. The PSB increases P availability to plants by solubilizing and mineralizing inorganic and organic soil phosphates. These bacteria solubilize mineral phosphate by producing organic acids [30,47] and thus lower the soil pH and mineralize organic phosphate by producing phosphatases [48]. In the present study, sole and co-inoculation combinations of PSB and ZSB strains promoted growth, yield, antioxidant activity, and nutrient uptake in cotton crops. Co-inoculation treatments showed more effective results than sole inoculations, which might be due to the growth-promoting abilities of inoculated ZSB and PSB strains, as confirmed earlier by in vitro characterization [49]. 

Results of present studies depicted that PSB and ZSB inoculation increased cotton growth, reproductive, and yield attributes compared to the absolute control and recommended NPK doses (Figure 1 and Figure 2; Table 2). Co-inoculation with *Bacillus subtilis* IA6 and *Bacillus* sp. IA16 proved more efficient in promoting growth, reproductive, and yield attributes than sole inoculations. The present study’s findings are similar to Hamed et al. [50], who studied the effects of *Bacillus circulance* and *Bacillus megatherium* at different fertilization levels [0%, 50%, and 100% recommended dose of fertilizers (RDF)]. They found more practical results at the 100% level of RDF. In contrast, minimum cotton growth and yield improvement occurred in the uninoculated control and without fertilizer. Qureshi et al. [51] reported an increase in shoot mass, root mass, shoot length, root length, boll weight, number of bolls, and seed cotton yield by applying *Rhizobium* isolate Br5.

Similarly, Arshad et al. [52] found that *Brevibacillus* sp. TN4-3NF significantly enhanced the lint and seed yield of transgenic and non-transgenic cotton plants compared with the uninoculated control plants. The improvement in growth, reproductive, and yield attributes by PSB and ZSB strains can be due to bacterial abilities to improve nutrient availability, phytohormones, and siderophores production, production of hydrolytic enzymes to combat pathogens, and most importantly, bacterial colonization to roots and thereby positive interaction with a plant [53,54].

Antioxidant enzyme systems neutralize the effect of free radicals on the cellular components of plants and are crucial for inducing immunity under harsh environments. Inoculation with a combination of PSB and ZSB strains significantly increased antioxidant activity compared to negative control and recommended NPK fertilizers (Figure 3). *Paenibacillus polymyxa* IA7 + *Bacillus aryabhattai* IA20 was significantly better in increasing CAT and SOD activity compared to control and recommended NPK fertilizers. Co-inoculation with *Paenibacillus polymyxa* IA7 + *Bacillus* sp. IA16 reported a significant rise in POX and GPX activity compared to the control and recommended NPK fertilizer. Similar to the results of our studies, the co-inoculation effect of bacterial strains under metal stress in plants showed more promising results, as reported by Fatnassi et al. [55]. They observed that co-inoculation reduced the deleterious effects of metal and enhanced CAT and SOD enzymes. In another study, Hahm et al. [56] reported an increase in enzymatic activities such as CAT, guaiacol peroxidase, and ascorbate peroxidase in pepper plants under salt stress after inoculation with three bacterial species, *Rhizobium massiliae, Brevibacterium iodinum*, and *Microbacterium oleivorans*. They concluded that plant growth-promoting bacteria decreased the hazardous effects of salts by increasing plant tolerance. Jha et al. [57] reported that bacterial strains *Pseudomonas pseudoalcaligenes* and *Bacillus pumilus* protect plants from salinity stress by increasing chlorophyll and antioxidant enzymes such as CAT and POX. Similarly, Singh et al. [58] used *Pseudomonas aeruginosa* for seed priming and reported that PGPB increased plant defense-related substances such as ascorbate peroxidase, superoxide dismutase, POX, phenylalanine ammonia-lyase in maize plants. 

Inoculation with bacterial strains enhances growth and growth-related attributes of cotton, owing to different direct and indirect growth-promoting traits. In present studies, co-inoculation with phosphate and Zn solubilizing strains enhanced growth and yield parameters of cotton crop that might be due to catalase activity, urease activity, phytohormone production, exopolysaccharides production [59] and siderophore production [60] by these strains that caused effective colonization of cotton roots by these strains leading to reach more nutrients (Zn, P) [61,62]. The plant growth-promoting abilities of *Bacillus* strains have already been reported in our previous study [28]. The results of our studies are supported by the work of Qureshi et al. [61], who reported an improvement in cotton growth due to the combined application of *Azotobacter* sp. and *Azospirillum* sp. at different levels of N and P. 

The inoculation of ZSB and PSB enhanced macro and micronutrient content in soil and their uptake in plant tissues (Appendix A, Table 3 and Table 4; Figure 4 and Figure 5) through solubilization, made it available for plants, increased root growth and resulted in several bolls, higher boll weight, and seed cotton yield. The combined application of ZSB and PSB led us toward their compatible nature and root colonization ability. They solubilized more nutrients and performed more functions according to their capabilities, which favored plant growth and yield attributes. Rafique et al. [62] characterized PGPB strains as promising multifarious mineral solubilizers, demonstrating an increase in quinoa’s growth and physiology. Li et al. [63] also reported that the combined application of mineral solubilizing PGPB strains promoted nutrient availability in soil and root growth of *Robinia pseudoacacia* through promoting mineral solubilization and enzymatic activities. Such bacterial strains adopt several mechanisms for improving nutrient availability in soil, such as P, K, Zn, and Fe solubilization [60,64,65,66,67]. The supplement of P fertilizer in soil for crops is insoluble rock phosphate fixed with Ca, Fe, and Al under certain environmental conditions [65]. The PGPB, particularly PSB, solubilizes phosphate by producing enzymes (phosphatase enzyme) and organic acids or H^+^ ions [46]. The PGPB converts insoluble forms of P and Zn into soluble and plant-usable forms by producing organic acids in the rhizosphere. These organic acids, including citric, acetic, lactic, gluconic, 2-keto-gluconic, malic, and oxalic acids, acidify the environment surrounding microbes [68]. This way, PGPB inoculation enhances soil fertility while reducing dependence on chemical fertilizers [69,70,71]. HCN production by PGPB also improves P availability by metal chelation and sequestration, as reported by Rijavec and Lapanje [72]. Similarly, PGPB produces siderophores for Fe availability and organic acids for Zn solubilization. The PGPB produced siderophores that bind to Fe^3+^ with high affinity. There is more attraction of bacterial siderophores toward Fe^3+^ than plants and fungi siderophores. The Fe from this complex is soluble and taken up by specific organisms [73].

## 5. Conclusions

Phosphate solubilizing and zinc solubilizing strains as sole and in a combined application can improve cotton growth, reproduction, antioxidant enzymatic activity, yield attributes, and nutrient uptake. Sole inoculation with *Bacillus aryabhattai* IA20 and *Paenibacillus polymyxa* IA7 increased growth, reproductive yield attributes, nutrient uptake, and antioxidant enzyme activities. However, the co-inoculation with *Bacillus subtilis* IA6 + *Bacillus* sp. IA16 and *Paenibacillus polymyxa* IA7 + *Bacillus aryabhattai* IA20 obtained the most significant improvement in these parameters. These findings conclude that combined PGPB applications may boost crop productivity through their multifunctional abilities. 

## Figures and Tables

**Figure 1 microorganisms-11-00861-f001:**
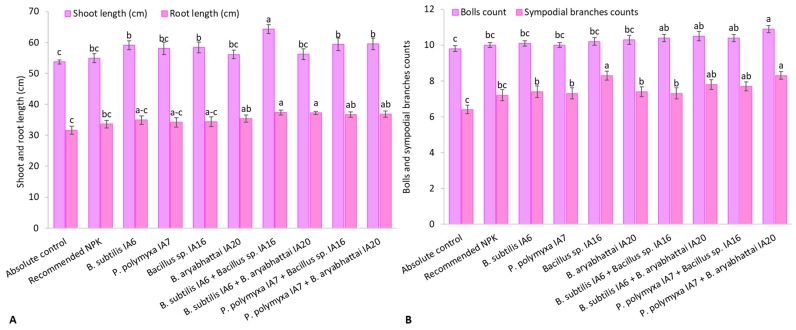
Effect of inoculation with PSB and ZSB strains on (**A**) shoot length and root length and (**B**) bolls counts and sympodial branches counts of cotton plants under pot trial; means sharing the same letter does not differ significantly.

**Figure 2 microorganisms-11-00861-f002:**
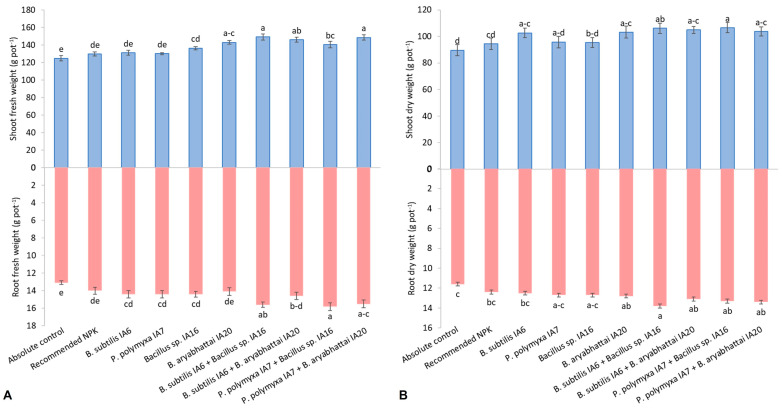
Effect of inoculation with PSB and ZSB strains on (**A**) fresh shoot and root weight and (**B**) dry shoot and root weight of cotton plants under pot trial; means sharing the same letter does not differ significantly.

**Figure 3 microorganisms-11-00861-f003:**
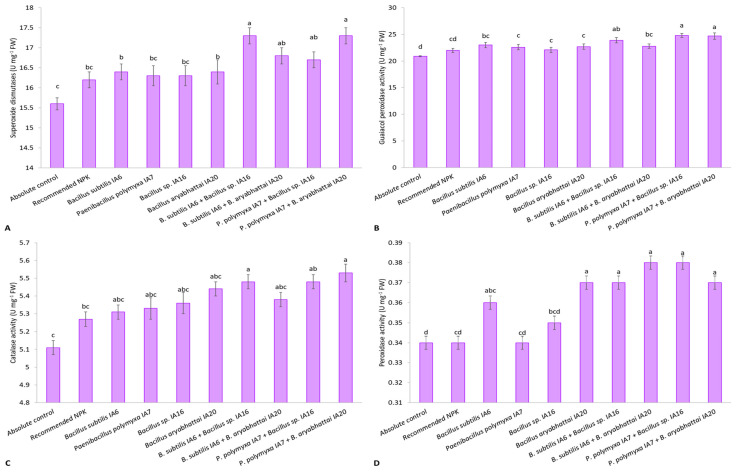
Effect of PSB and ZSB inoculation on superoxide dismutase (**A**), guaiacol peroxidase (**B**), catalase (**C**), and peroxidase (**D**) activities of cotton leaves under pot trial; means sharing the same letter does not differ significantly.

**Figure 4 microorganisms-11-00861-f004:**
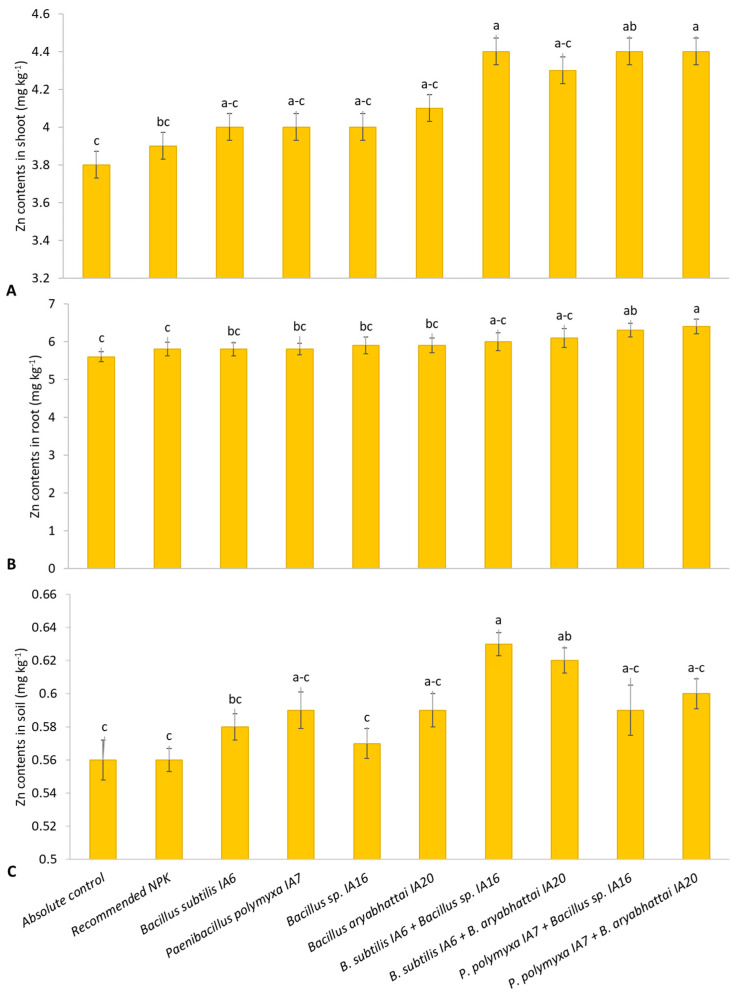
Effect of PSB and ZSB inoculation Zn content in shoot (**A**) root (**B**), and soil (**C**) of cotton plants under pot trial; means sharing the same letter does not differ significantly.

**Figure 5 microorganisms-11-00861-f005:**
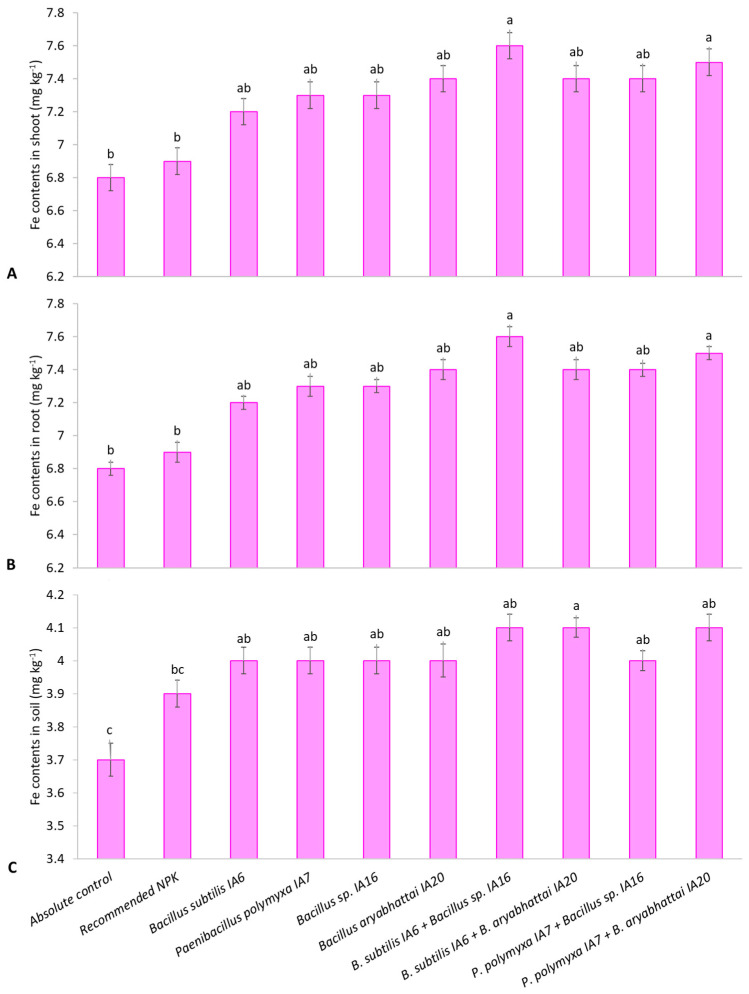
Effect of PSB and ZSB inoculation Fe content in shoot (**A**), root (**B**), and soil (**C**) of cotton plants under pot trial; means sharing the same letter does not differ significantly.

**Table 1 microorganisms-11-00861-t001:** Physicochemical characteristics of the soil used in a pot trial.

Characteristic	Value
Sand	43%
Silt	43%
Clay	14%
Textural class	Loamy soil
Saturation percentage	37.0%
pHs	7.9
ECe	1.63 dS m^−1^
Organic matter	0.28%
Available phosphorus	5.1 mg kg^−1^
Extractable potassium	79 mg kg^−1^
Total nitrogen	0.023%
Available zinc	0.65 mg kg^−1^
Available iron	3.7 mg kg^−1^

**Table 2 microorganisms-11-00861-t002:** Effect of PSB and ZSB strains inoculation on cotton boll weight, seed yield, lint yield, and lint percentage; means sharing the same letter does not differ significantly.

Treatment	Single Boll Weight (g)	Seed Cotton Yield (g pot^−1^)	Lint Yield (g pot^−1^)	Lint Percentage (%)
Absolute control	2.5 ± 0.03 e	49.1 ± 1.28 e	15.1 ± 0.52 e	30.8 ± 1.13 a
Recommended NPK	2.7 ± 0.03 d	53.4 ± 1.08 d	16.8 ± 0.48 d	30.9 ± 0.91 a
*B. subtilis* IA6	2.8 ± 0.02 bc	55.6 ± 1.14 cd	17.6 ± 0.55 b–d	31.0 ± 1.18 a
*P. polymyxa* IA7	2.7 ± 0.02 cd	54.4 ± 0.82 cd	17.1 ± 0.25 cd	31.3 ± 0.27 a
*Bacillus* sp. IA16	2.8 ± 0.03 bc	56.3 ± 1.58 b–d	17.5 ± 0.22 b–d	30.9 ± 0.57 a
*B. aryabhattai* IA20	2.8 ± 0.02 a–c	57.3 ± 1.27 a–c	18.1 ± 0.27 a–c	31.1 ± 0.90 a
*B. subtilis* IA6 + *Bacillus* sp. IA16	2.8 ± 0.04 a–c	58.2 ± 1.49 a–c	17.8 ± 0.54 a–d	31.0 ± 1.32 a
*B. subtilis* IA6 + *B. aryabhattai* IA20	2.8 ± 0.02 ab	59.5 ± 1.79 ab	18.7 ± 0.33 ab	31.0 ± 1.42 a
*P. polymyxa* IA7 + *Bacillus* sp. IA16	2.9 ± 0.04 a	59.5 ± 1.79 ab	18.5 ± 0.50 ab	31.5 ± 1.36 a
*P. polymyxa* IA7 + *B. aryabhattai* IA20	2.8 ± 0.03 a–c	60.9 ± 0.67 a	18.8 ± 0.31 a	31.2 ± 0.57 a
LSD (*p* ≤ 0.05)	0.0821	3.8061	1.1868	2.9277

**Table 3 microorganisms-11-00861-t003:** Effect of PSB and ZSB strains inoculation on N, P, and K contents in root of cotton; means sharing the same letter does not differ significantly.

Treatment	N content in Root (%)	P content in Root (%)	K content in Root (%)
Absolute control	2.15 ± 0.008 g	0.39 ± 0.007 d	1.34 ± 0.005 e
Recommended NPK	2.23 ± 0.003 f	0.41 ± 0.007 cd	1.37 ± 0.004 d
*B. subtilis* IA6	2.24 ± 0.007 ef	0.42 ± 0.009 a–c	1.38 ± 0.003 d
*P. polymyxa* IA7	2.24 ± 0.007 ef	0.43 ± 0.008 a–c	1.39 ± 0.006 d
*Bacillus* sp. IA16	2.27 ± 0.008 d	0.41 ± 0.006 bc	1.41 ± 0.018 c
*B. aryabhattai* IA20	2.25 ± 0.006 e	0.42 ± 0.009 a–c	1.44 ± 0.012 b
*B. subtilis* IA6 + *Bacillus* sp. IA16	2.34 ± 0.006 a	0.43 ± 0.005 ab	1.46 ± 0.007 ab
*B. subtilis* IA6 + *B. aryabhattai* IA20	2.31 ± 0.007 c	0.42 ± 0.008 a–c	1.45 ± 0.009 ab
*P. polymyxa* IA7 + *Bacillus* sp. IA16	2.32 ± 0.003 bc	0.43 ± 0.006 a–c	1.47 ± 0.008 a
*P. polymyxa* IA7 + *B. aryabhattai* IA20	2.33 ± 0.004 ab	0.44 ± 0.008 a	1.46 ± 0.008 ab
LSD (*p* ≤ 0.05)	0.0170	0.0209	0.0256

**Table 4 microorganisms-11-00861-t004:** Effect of PSB and ZSB strains inoculation on N, P, and K contents in a shoot of cotton; means sharing the same letter does not differ significantly.

Treatment	N Content in Shoot (%)	P Content in Shoot (%)	K Content in Shoot (%)
Absolute control	2.12 ± 0.006 f	0.29 ± 0.007 b	1.32 ± 0.006 d
Recommended NPK	2.20 ± 0.009 e	0.30 ± 0.005 ab	1.37 ± 0.005 c
*B. subtilis* IA6	2.22 ± 0.010 de	0.31 ± 0.008 ab	1.37 ± 0.004 c
*P. polymyxa* IA7	2.21 ± 0.010 de	0.32 ± 0.008 a	1.38 ± 0.007 c
*Bacillus* sp. IA16	2.21 ± 0.008 de	0.30 ± 0.004 ab	1.38 ± 0.006 c
*B. aryabhattai* IA20	2.22 ± 0.007 de	0.31 ± 0.009 ab	1.42 ± 0.008 b
*B. subtilis* IA6 + *Bacillus* sp. IA16	2.29 ± 0.012 a	0.32 ± 0.008 a	1.43 ± 0.008 ab
*B. subtilis* IA6 + *B. aryabhattai* IA20	2.25 ± 0.006 bc	0.32 ± 0.008 a	1.43 ± 0.006 b
*P. polymyxa* IA7 + *Bacillus* sp. IA16	2.23 ± 0.007 cd	0.32 ± 0.009 a	1.45 ± 0.006 a
*P. polymyxa* IA7 + *B. aryabhattai* IA20	2.27 ± 0.005 ab	0.32 ± 0.009 a	1.44 ± 0.007 ab
LSD (*p* ≤ 0.05)	0.0234	0.0220	0.0178

## Data Availability

Not applicable.

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
