# Peer review of "Mineral-Solubilizing Bacteria-Mediated Enzymatic Regulation and Nutrient Acquisition Benefit Cotton’s (Gossypium hirsutum L.) Vegetative and Reproductive Growth"

_microorganisms, 2023, doi:10.3390/microorganisms11040861_

Round 1

Reviewer 1 Report

.I accept in present form

Author Response

Respected Reviewer, 

Thank you for evaluating our manuscript. 

Warmed regards, 

Muhammad Zahid Mumtaz

Reviewer 2 Report

Dear Authors,

Thank you for the opportunity to review your manuscript “Mineral-solubilizing bacteria-mediated enzymatic regulation and nutrient acquisition confer vegetative and reproductive growth of cotton (Gossypium hirsutum L.)”. This is straightforward and very thoroughly conducted research, revealing the importance of phosphate and zinc solubilizing bacterial endosymbionts in the nutrition of cotton plants, which belongs to the major agricultural crops of our planet and is a valuable commodity of Pakistan’s agriculture.

I don’t have any critic or suggestions regarding the experiment’s design, methodology or the obtained outcomes. It is a well-thought, solid and elegant investigation done with sufficient statistical support and undisputable results. However, solely from the readers perspective I found a few spots where some clarification would be beneficial, I believe.

Title. Would verb “benefit” describe the effect of used bacteria better, rather than “confer”?

NPK needs to be explained by its first mentioning.

Line 87. I would change the verb “collected” to “obtained” or “acquired.” The reason is that you use this verb downstream in slightly different meaning. Although grammatically correct, it might be confusing.

Line 91. As I understand, bacterial inoculum coats not only the seeds, but also the carrier material, which is used for seeds planting, and there is no way to correctly determine the actual contribution of seeds coating to the seedling’s nutrition, compared to the carrier slur? Is this a common practice to sow cotton seeds together with some mineral carrier?

Line 112. How did you maintain the 1:1 ratio of the bacterial inoculants, using cell count?

Line 149. What does it mean, “plant-1”? Is this the number of bolls per plant?

Line 189. “respectively, respectively…”

Lines 207, 214, 222. Would be not better to use the verb “increased’ instead of “improved”?

However, my few remarks don’t diminish the value of the research presented by the authors. I hope to see this manuscript published soon on the pages of the Microorganisms journal.

Sincerely,

Reviewer.

Author Response

Response to comments

Point 1: Thank you for the opportunity to review your manuscript “Mineral-solubilizing bacteria-mediated enzymatic regulation and nutrient acquisition confer vegetative and reproductive growth of cotton (Gossypium hirsutum L.)”. This is straightforward and very thoroughly conducted research, revealing the importance of phosphate and zinc solubilizing bacterial endosymbionts in the nutrition of cotton plants, which belongs to the major agricultural crops of our planet and is a valuable commodity of Pakistan’s agriculture.

Response: Thank you so much for appreciating our work.

Point 2: I don’t have any critic or suggestions regarding the experiment’s design, methodology or the obtained outcomes. It is a well-thought, solid and elegant investigation done with sufficient statistical support and undisputable results. However, solely from the readers perspective I found a few spots where some clarification would be beneficial, I believe.

Response: Thank you for providing valuable comments. We have revised the manuscript as per your suggestion.

Point 3: Title. Would verb “benefit” describe the effect of used bacteria better, rather than “confer”?

Response: The verb “confer” is replaced with “benefit” and a slight change in the manuscript title is also made to correct the title's syntax. The new manuscript title is “Mineral-solubilizing bacteria-mediated enzymatic regulation and nutrient acquisition benefit cotton’s (Gossypium hirsutum L.) vegetative and reproductive growth”

Point 4: NPK needs to be explained by its first mentioning.

Response: We mentioned the full form of NPK in the revised version.

Point 5: Line 87. I would change the verb “collected” to “obtained” or “acquired.” The reason is that you use this verb downstream in slightly different meaning. Although grammatically correct, it might be confusing.

Response: Corrected

Point 6: Line 91. As I understand, bacterial inoculum coats not only the seeds, but also the carrier material, which is used for seeds planting, and there is no way to correctly determine the actual contribution of seeds coating to the seedling’s nutrition, compared to the carrier slur? Is this a common practice to sow cotton seeds together with some mineral carrier?

Response: Our primary focus is to deliver the bacterial inoculant (increased population) in the root zone of cotton that’s why inoculum was coated on the seed along with carrier material. The carrier material clay and sugar are used to obtain a firmly fixed coating of bioinoculant on cotton. While peat (organic matter) in seed coating helps to improve the viability of bacterial inoculants in the soil environment. So our ultimate target is to enhance the delivery of bacterial inoculants in the root zone of cotton through the coating on seeds (peat was part of the coating both in treated plants and the control group). The application of organic matter (peat) is essential to improve the viability of bacterial inoculants, as our experimental soil is already organic matter deficient (0.28%).

Point 7: Line 112. How did you maintain the 1:1 ratio of the bacterial inoculants, using cell count?

Response: Strain broth cultures were maintained at a 1:1 ratio by applying bacterial liquid cultures of 0.70 optical density at 600 nm, ultimately based on cell count. This statement is provided in the revised manuscript.

Point 8: Line 149. What does it mean, “plant-1”? Is this the number of bolls per plant?

Response: Yes, “plant-1” is meant for bolls per plant. Correct in the revised manuscript.

Point 9: Line 189. “respectively, respectively…”

Response: Corrected

Point 10: Lines 207, 214, 222. Would be not better to use the verb “increased’ instead of “improved”?

Response: The verb “improved” is replaced with “increased” throughout the revised manuscript.  

Point 11: However, my few remarks don’t diminish the value of the research presented by the authors. I hope to see this manuscript published soon on the pages of the Microorganisms journal.

Response: Thank you for evaluating our manuscript.

Reviewer 3 Report

The present paper deals with an interesting topic but needs improvements before it can be published. In my point of view, it cannot be published in the present version.

Materials and methods are presented as a list of the activities performed, leaving out all the details related to the protocols. The authors, while reporting reference works, are invited to describe (even briefly) the activities carried out to allow not only an evaluation but also the repeatability of the evidence presented.

In the results section, the authors present a series of tables that are often difficult to follow because they are full of information. I suggest better organizing the results with graphs and, given the number of results, moving some complementary information to the supplementary materials section. Pictures are welcome.

I suggest creating a more homogeneous and not separate discussion.

The bibliography needs careful revision in line with the standards of the journal as in some cases fundamental details for the search are missing

Overall, I recommend a thorough review of the grammar and syntax.

Author Response

Response to comments

Point 1: The present paper deals with an interesting topic but needs improvements before it can be published. In my point of view, it cannot be published in the present version.

Response 1: Thank you for evaluating our manuscript. We have revised the manuscript as per your suggestion.      

Point 2: Materials and methods are presented as a list of the activities performed, leaving out all the details related to the protocols. The authors, while reporting reference works, are invited to describe (even briefly) the activities carried out to allow not only an evaluation but also the repeatability of the evidence presented.

Response 2: The detailed protocol of all the referenced activities is fully explored in the revised manuscript highlighted in red.  

Point 3: In the results section, the authors present a series of tables that are often difficult to follow because they are full of information. I suggest better organizing the results with graphs and, given the number of results, moving some complementary information to the supplementary materials section. Pictures are welcome.

Response 3: We have reduced the number of tables by providing the data in figure form. Some data in the table data are also moved into the supplementary material.

Point 4: I suggest creating a more homogeneous and not separate discussion.

Response 4: the discussion section is revised, and homogeneous discussion is provided by removing separate headings of observations.

Point 5: The bibliography needs careful revision in line with the standards of the journal as in some cases fundamental details for the search are missing

Response 5: The bibliography is carefully revised per journal standards style and formatting.  

Point 6: Overall, I recommend a thorough review of the grammar and syntax

Response 6: Thank you for the suggestion. Our English native co-author Lisa Pataczek thoroughly revises English of this manuscript  

Round 2

Reviewer 3 Report

In the Introduction section, I would add some examples of the application of PGPRs with the same purpose you propose. I also suggest summarizing in more words the content and objective of the research.

L71: Plant growth-promoting bacteria are PGPB. Substitute PGPB throughout the article...

Ls 84-92: what did you mean by DF minimal broth? Report a reference or if you buy it, the manufacturer’s address. Have the strains used ever been tested previously for the same purpose? How did you test the OD?

Ls 94-124: I appreciate the integration reported by the authors. I suggest one more effort in order to rephrase the paragraph in a clearer order (i.e., try to divide the section into paragraphs according to the method described).

-          Where did you take the soil used?

-          Bouyoucos hydrometer. Report manufacturer details.

-          Soil texture triangle. Report a reference.

-          Specify from the beginning of the sentence that the saturated mixture was used to calculate the pH.

-          Calibrated digital pH meter. Report manufacturer details.

-          Kjeldahl apparatus. Report manufacturer details.

-          Few pumice-boiling granules. What do you mean by “few”? Also, add details about the manufacturer.

-          “The digest was cooled, filtered, and diluted up to 250 mL”. How did you cool it? How did you filter it? How did you dilute it?

-          Flame photometer. Report manufacturer details.

-          “The physicochemical properties of the soil used in a pot trial are given in Table 1”. This table should be reported in the result section.

-          And so on…

Ls 126-147: Mention the protocol followed for the inoculation of the PGPB. How did you obtain your coated seeds? What do you mean by “recommended” and “not-recommended doses”? Which kind of pots did you use (material and volume)? Does the absolute control correspond to no-treated plants?

                “air-dried and sieved soil”. How did you obtain these parameters?

                “natural conditions”. What do you mean?

                “Antioxidant activities were estimated at the flowering stage. The growth and yield parameters were determined at harvesting. After harvesting the crop, soil, and plant samples were collected to assess plant nutrient uptake.” Can you report approximately how many days post inoculation (or transplanting) did you perform this analysis?

L153: How did you homogenize? Which instrument did you use?

Ls149-172: I appreciate the integration reported by the authors. I suggest one more effort in order to rephrase the paragraph in a clearer order (i.e., try to divide the section into paragraphs according to the method described).

Ls 174-178: How did you obtain the dry weights? In general, how did you measure every single parameter?

Ls 180-194: I appreciate the integration reported by the authors. I suggest one more effort in order to rephrase the paragraph in a clearer order (i.e., try to divide the section into paragraphs according to the method described).

Regarding the graphs proposed, I really like the colorful effect given, but I suggest using a greyscale to facilitate the reader evaluation.

Ls 483-486: According to your results, this is a crucial point. I suggest an integration with other practical examples of the use of bacterial consortia with the same purpose you suggest in your research.

Author Response

Response to comments

In the Introduction section, I would add some examples of the application of PGPRs with the same purpose you propose. I also suggest summarizing in more words the content and objective of the research.

Response: examples are provided. The content and objective of the research are elaborated briefly in the revised version of the manuscript. 

L71: Plant growth-promoting bacteria are PGPB. Substitute PGPB throughout the article...

Response: Corrected.

Ls 84-92: what did you mean by DF minimal broth? Report a reference or if you buy it, the manufacturer’s address. Have the strains used ever been tested previously for the same purpose? How did you test the OD?

Response: DF stands for Dworkin and Foster (DF) salt minimal broth, and its full form is provided in the revised version of the manuscript, along with its reference. These strains were previously tested for plant growth promotion under control conditions and in vitro plant growth-promoting characteristics, and the OD was tested at 600 nm wavelength. All this required information is provided in the revised version of the manuscript as per your suggestion.   

Ls 94-124: I appreciate the integration reported by the authors. I suggest one more effort in order to rephrase the paragraph in a clearer order (i.e., try to divide the section into paragraphs according to the method described).

Response: corrected as per your suggestion.

-          Where did you take the soil used?

Response: Information is provided

-          Bouyoucos hydrometer. Report manufacturer details.

Response: details are provided

-          Soil texture triangle. Report a reference.

Response: Reference is provided

-          Specify from the beginning of the sentence that the saturated mixture was used to calculate the pH.

Response: Information is provided

-          Calibrated digital pH meter. Report manufacturer details.

Response: detail is provided

-          Kjeldahl apparatus. Report manufacturer details.

Response: detail is provided

-          Few pumice-boiling granules. What do you mean by “few”? Also, add details about the manufacturer.

Response: Information is provided

- “The digest was cooled, filtered, and diluted up to 250 mL”. How did you cool it? How did you filter it? How did you dilute it?

Response: detail is provided

-          Flame photometer. Report manufacturer details.

Response: detail is provided

-          “The physicochemical properties of the soil used in a pot trial are given in Table 1”. This table should be reported in the result section.

Response: Respected reviewer, we did not discuss the physicochemical properties in the results and discussion sections. The result section starts directly with a description of shoot and root growth. It will be better to avoid mentioning the soil physicochemical information in the results because it is unrelated to any outcomes. We publish soil physicochemical properties almost in every research paper, and multiple reviewers have different points of view. In conclusion, we decided to keep soil physicochemical properties in the methodology section with due respect.   

Ls 126-147: Mention the protocol followed for the inoculation of the PGPB. How did you obtain your coated seeds? What do you mean by “recommended” and “not-recommended doses”? Which kind of pots did you use (material and volume)? Does the absolute control correspond to no-treated plants?

Response: The inoculation information is provided in L107-109. However, we provided these details as per your suggestion. The recommended doses of NPK are meant the quantities of required NPK nutrients suggested by the Department of Agriculture, Punjab, Pakistan. Such information was already given in the below line. However, we have moved the information in the mentioned section as per your suggestion.  

                “air-dried and sieved soil”. How did you obtain these parameters?

Response: The inoculation information is provided in L107-109. However, we provided these details as per your suggestion. Air-dried soil is an obvious message; however, we have tried to make a clear understanding. We mentioned it as air-dried under shade for two weeks. All the parameters procedure is already provided in depth.

                “natural conditions”. What do you mean?

Response: Its natural climatic conditions.

                “Antioxidant activities were estimated at the flowering stage. The growth and yield parameters were determined at harvesting. After harvesting the crop, soil, and plant samples were collected to assess plant nutrient uptake.” Can you report approximately how many days post inoculation (or transplanting) did you perform this analysis?

Response: We apologize; at this time, we do not have the exact number of days.

L153: How did you homogenize? Which instrument did you use?

Response: homogenized through a vortex.

Ls149-172: I appreciate the integration reported by the authors. I suggest one more effort in order to rephrase the paragraph in a clearer order (i.e., try to divide the section into paragraphs according to the method described).

Response: Done

Ls 174-178: How did you obtain the dry weights? In general, how did you measure every single parameter?

Response: the required information is provided.

Ls 180-194: I appreciate the integration reported by the authors. I suggest one more effort in order to rephrase the paragraph in a clearer order (i.e., try to divide the section into paragraphs according to the method described).

Response: revised as per your suggestion.

Regarding the graphs proposed, I really like the colorful effect given, but I suggest using a greyscale to facilitate the reader evaluation.

Response: We admire your suggestion; however, we love to present these graphs in fascinating colors. During revision, these graphs took almost a whole day to plan and finalize a single graph. I request you to retain these colorful graphs as microorganisms journal publishes articles online that will fascinate the findings.  

Ls 483-486: According to your results, this is a crucial point. I suggest an integration with other practical examples of the use of bacterial consortia with the same purpose you suggest in your research.

Response: examples are provided as per your suggestion.
